# Metabolome and Transcriptome Joint Analysis Reveals That Different Sucrose Levels Regulate the Production of Flavonoids and Stilbenes in Grape Callus Culture

**DOI:** 10.3390/ijms251910398

**Published:** 2024-09-27

**Authors:** Xiaojiao Gu, Zhiyi Fan, Yuan Wang, Jiajun He, Chuanlin Zheng, Huiqin Ma

**Affiliations:** College of Horticulture, China Agricultural University, Beijing 100083, China; guxj0906@163.com (X.G.); b20213170865@cau.edu.cn (Z.F.); s20213172748@cau.edu.cn (Y.W.); h374@163.com (J.H.)

**Keywords:** grape callus, flavonoids, stilbenes, metabolome, transcriptome, sucrose concentration

## Abstract

To reveal the effect of sucrose concentration on the production of secondary metabolites, a metabolome and transcriptome joint analysis was carried out using callus induced from grape variety Mio Red cambial meristematic cells. We identified 559 metabolites—mainly flavonoids, phenolic acids, and stilbenoids—as differential content metabolites (fold change ≥2 or ≤0.5) in at least one pairwise comparison of treatments with 7.5, 15, or 30 g/L sucrose in the growing media for 15 or 30 days (d). Resveratrol, viniferin, and amurensin contents were highest at 15 d of subculture; piceid, ampelopsin, and pterostilbene had higher contents at 30 d. A transcriptome analysis identified 1310 and 498 (at 15 d) and 1696 and 2211 (at 30 d) differentially expressed genes (DEGs; log2(fold change) ≥ 1, *p* < 0.05) in 7.5 vs. 15 g/L and 15 vs. 30 g/L sucrose treatments, respectively. In phenylpropane and isoflavone pathways, DEGs encoding cinnamic acid 4-hydroxylase, chalcone synthase, chalcone isomerase, and flavanone 3-hydroxylase were more highly expressed at 15 d than at 30 d, while other DEGs showed different regulation patterns corresponding to sucrose concentrations and cultivation times. For all three sucrose concentrations, the stilbene synthase (*STS*) gene exhibited significantly higher expression at 15 vs. 30 d, while two resveratrol O-methyltransferase (*ROMT*) genes related to pterostilbene synthesis showed significantly higher expression at 30 vs. 15 d. In addition, a total of 481 DEGs were annotated as transcription factors in pairwise comparisons; an integrative analysis suggested MYB59, WRKY20, and MADS8 as potential regulators responding to sucrose levels in flavonoid and stilbene biosynthesis in grape callus. Our results provide valuable information for high-efficiency production of flavonoids and stilbenes using grape callus.

## 1. Introduction

Flavonoids are plant secondary metabolites that include major subclasses of chalcones, flavones, flavonols, and isoflavones. They can be modified by glycosylation, methylation, and acylation to produce a variety of derivatives [1,2]. Stilbenes, the best known being resveratrol, are natural phytoalexins found in plants [3]. Studies have demonstrated that flavonoids and stilbenes possess antioxidant, anticancer, antibacterial, and anti-inflammatory properties, among other health benefits. They are widely used in medicine and cosmetics [4,5,6,7].

The biosynthesis of both flavonoids and stilbenes starts from the phenylpropane pathway [8] using phenylalanine as the substrate, and 4-coumaroyl-CoA is synthesized under the co-catalysis of phenylalanine ammonia lyase (PAL), cinnamic acid 4-hydroxylase (C4H), and 4-coumarate-CoA ligase (4CL) [9]. The synthesis of chalcone and resveratrol is then catalyzed by chalcone synthase and STS using 4-coumaroyl-CoA and malonyl-CoA as substrates, respectively. Various flavonoids are synthesized from chalcone via catalysis by enzymes such as chalcone isomerase (CHI), flavanone 3-hydroxylase (F3H), and F3′H [8]; catalysis of resveratrol by the corresponding enzymes can generate some biologically active derivatives, such as glycosylation to produce resveratrol glycosides, hydroxylation to produce leupeptin, and dimerization to produce ε-viniferin; resveratrol can also be methoxylated by ROMT to produce pterostilbene [10,11,12,13,14].

Grapevine is well known for its high content of flavonoids and stilbenes. Specific contents of these compounds in berries, leaves, shoots, and other organs may vary among grape varieties and the condition of the tissue [15]. Although flavonoids and stilbenes can be extracted from plant tissues, including grapevine, at specific growth stages, the composition and yield are highly influenced by environmental factors and harvesting conditions [16]. Plant tissue culture technology is based on the inherent biosynthetic pathways of plant cells or tissues, utilizing endogenous enzyme systems to produce specific compounds or increase the yield of desired compounds by adjusting the culture conditions [17]. This technology is highly useful to the pharmaceutical, food, and cosmetic industries.

Carbon sources play a crucial role in the growth of plant callus, and sucrose is the standard carbon source for inducing, proliferating, and differentiating various types of plant callus [18]. Sucrose not only provides an energy source for the callus; it also regulates the osmotic pressure in the culture medium. The sucrose content has a significant impact on both the growth of grape callus cells and the accumulation of secondary metabolites. Related research has shown that an increase in sucrose concentration from 0 to 60 mM can increase the expression of the transcription factor (TF) MYB75, thereby regulating the expression of anthocyanin-synthesis genes and leading to significant accumulation of plant anthocyanins [19,20]. In grape var. Barbera suspension cells, an increase in sucrose concentration from 10 to 30 or 40 g/L was observed to promote cell growth and phenylpropanoid biosynthesis. Peak stilbene content was detected on day 2 in the 40 g/L sucrose-treated cultures compared to cultivation with 10 and 30 g/L sucrose; the polyphenol content increased 2.0 and 1.8 times in the 40 g/L sucrose-treated cultures, respectively, and the resveratrol content increased 3.5-fold [21].

In this study, we elucidate the relationship between the accumulation of flavonoids and stilbenes in grape callus cells and different sucrose contents and cultivation times by combining metabolomics and transcriptomics analyses. Potential key regulatory factors involved in the regulation of flavonoid and stilbene accumulation are revealed, and we provide a theoretical basis for the accumulation and utilization of flavonoids, resveratrol, and their derivatives in grape callus cells.

## 2. Results

### 2.1. Phenotype and Proliferation Rate of Grape Callus under Different Sucrose Concentrations and Cultivation Times

On media with 7.5 g/L and 15 g/L sucrose (Suc7.5 and Suc15, respectively), the callus appeared pale yellow in color; on medium with 30 g/L sucrose (Suc30), partial milky white fluffy callus appeared from day (d) 15 of subculture (T1) onwards (Figure 1A). After 15 d of culture, the fresh weight of the callus in Suc15-T1 increased rapidly, with a growth rate of 352.1%, whereas the fresh weights of the callus tissues in Suc30-T1 and Suc7.5-T1 increased more slowly, with growth rates of 221.3% and 99.6%, respectively. After 30 d of culture (T2), there was no significant difference in callus fresh weight between Suc15-T2 and Suc30-T2, with growth rates of 533.3% and 730.3%, respectively. The fresh weight of the Suc7.5-T2 callus tissue was significantly lower than that of Suc15-T2 and Suc30-T2, with a growth rate of 325.7% (Figure 1B). These results indicated that the low and high sucrose contents in the medium might limit grape tissue growth due to the callus being in a state of carbon/sucrose source stress. When the sucrose content was 15 g/L, the callus grew faster after 15 d of culture with no production of white, fluffy, non-embryonic callus.

### 2.2. Identification of Metabolites with Differential Contents 

A total of 559 metabolites with differential contents (differential content metabolites [DCMs]) were identified in the grape callus by comprehensive targeted metabolomic analysis, including 13 categories. Among them, amino acids and derivatives and phenolic acids contained a relatively large number of metabolites (Figure 2A, Appendix A). A principal component analysis (PCA) (Figure 2B) and a hierarchical clustering analysis (Appendix A) showed that the three biological replicates of each sample cluster together, indicating good repeatability and reliability of the metabolomics data. Based on these results, the six groups of samples could be clearly divided into four categories, with Suc7.5-T1 and Suc15-T1 clustering together, Suc7.5-T2 and Suc15-T2 clustering together, and Suc30-T1 and Suc30-T2 each forming an independent category (Figure 2B). Overall, Suc7.5 and Suc15 were closer in terms of metabolic features.

Pairwise differential metabolite comparisons were conducted for the callus materials at two time points with three different sucrose concentrations. The comparison between Suc30-T2 and Suc15-T2 had the highest number of DCMs, with 347, whereas the comparison between Suc15-T2 and Suc7.5-T2 had the lowest number, with 159 (Figure 2C and Appendix A). A Venn diagram of the T1 groups showed 77 common DCMs (Figure 2D), while the T2 groups had 94 common DCMs (Figure 2E); the T1 groups vs. T2 groups showed 67 common DCMs (Figure 2E). Among the identified DCMs, flavonoids represented the most abundant category in both T1 and T2 groups, accounting for 34.33% and 36.36%, respectively (Appendix A). Flavonoids and stilbenes were the most abundant common DCMs.

### 2.3. Differential Accumulation of Flavonoids and Stilbenes in Grape Callus

A total of 94 flavonoid metabolites and 28 stilbenes were detected in all samples. The flavonoid metabolites included 4 chalcones, 7 flavanols, 6 flavanonols, 17 flavanones, 21 flavones, and 39 flavonols (Appendix A). A cluster heatmap analysis of the concentrations of the 94 flavonoid metabolites revealed three distinct accumulation patterns: the first type of DCMs exhibited high accumulation in Suc7.5-T1 and Suc15-T1 (Figure 3A), the second type exhibited high accumulation in Suc7.5-T2 and Suc15-T2 (Figure 3B), and the third type exhibited high accumulation in Suc30-T2 (Figure 3C). Similarly, a cluster heatmap analysis of the 28 stilbene compounds revealed three distinct accumulation patterns: the first type exhibited high accumulation in Suc7.5-T1, Suc15-T1, and Suc30-T1, the second exhibited high accumulation in Suc30-T2, and the third exhibited high accumulation in Suc7.5-T2 and Suc15-T2 (Figure 3D).

### 2.4. Identification of DEGs and Enrichment Analysis

The transcriptome analysis was conducted on grape callus grown with different sucrose concentrations and cultivation times. The PCA and hierarchical clustering plots showed that each group of three biological replicates clustered closely together, with clear separation between the groups (Appendix A). Comparing sucrose concentrations in the culture medium, the number of upregulated genes was significantly higher than the number of downregulated genes with increasing sugar concentration at 15 d of callus cultivation. There were 908 upregulated and 402 downregulated genes in Suc15-T1 vs. Suc7.5-T1 and 379 upregulated and 119 downregulated genes in Suc30-T1 vs. Suc15-T1. After 30 d of callus cultivation, there were 587 upregulated and 1109 downregulated genes in Suc15-T2 vs. Suc7.5-T2 and 1303 upregulated and 908 downregulated genes in Suc30-T2 vs. Suc15-T2. Comparing the effect of cultivation time under the same sucrose concentration conditions, the largest number of DEGs was observed in Suc7.5-T1 vs. Suc7.5-T2, with 1512 upregulated and 2311 downregulated DEGs, while the smallest number was in Suc30-T1 vs. Suc30-T2, with 181 upregulated and 90 downregulated DEGs. In Suc15-T1 vs. Suc15-T2, there were 1296 upregulated and 1293 downregulated DEGs (Figure 4A and Appendix A).

There were 73 (Figure 4C), 244 (Figure 4D), and 61 (Figure 4E) common DEGs in T1, T2, and T1 vs. T2, respectively. To further evaluate the biological functions of the common DEGs in T1 vs. T2, a KEGG enrichment analysis was performed. The results revealed significant enrichment in the following pathways: phenylalanine metabolism (ko00940) and stilbenoid, diarylheptanoid, and gingerol biosynthesis (ko00945) (Figure 4F, Appendix A). Comparison of the T2 groups showed the highest number of DEGs, with significant enrichment in pathways involved in stilbenoid, diarylheptanoid, and gingerol biosynthesis (Appendix A). Prolonged carbon source deficiency stress caused significant changes in the expression of a large number of genes in grape callus, consistent with the metabolomic results, indicating that the changes in metabolite accumulation were controlled by differential gene expression. A focused analysis was conducted on gene-expression regulation related to sugar absorption and metabolism as well as flavonoid and stilbene biosynthesis.

### 2.5. DEGs in the Sucrose Metabolism Pathway

We found 17 DEGs related to sucrose metabolism in grape callus, including invertase, sucrose synthase, and sugar transporter proteins, among others. There were four invertase genes: *VINV*, *VINV1*, *NINVC*, and *NINVE*. Their expression levels decreased with increasing sucrose concentration in the culture medium but increased with cultivation time. It is worth noting that the expression pattern of two sucrose synthase genes, *SUS* and *SUS2*, was opposite to that of the invertase genes, namely, their expression levels increased with sucrose concentration and gradually decreased with cultivation time. The genes *SWEET2*, *SWEET16*, and *SWEET2a* showed a positive correlation with sucrose concentration in the medium at 15 d, whereas *ERD6-like 16*, *ERD6-like 7*, and *ERD6-like 6* showed significant differences at 30 d, decreasing with increasing sugar content. *HT*, *HT5*, *SUC11*, and *SUC27* exhibited the highest expression levels at sucrose concentrations of 7.5 g/L and 15 g/L (Figure 4B, Appendix A).

### 2.6. Metabolism and Gene Co-Expression Networks in Flavonoid and Stilbene Biosynthesis

To understand the network regulating the changes in the content of flavonoids and stilbenes in grape callus with different sucrose concentrations and cultivation times, we integrated the metabolomic and transcriptomic data through weighted gene co-expression network analysis (WGCNA). Based on the co-expression patterns, 21 distinct gene-expression modules were identified. These modules were marked in different colors and presented in the form of a heatmap and clustering diagram (Appendix A). We used the phenotypic data of 12 flavonoids and 7 stilbenes with antioxidant and anti-inflammatory activities for a module–trait correlation analysis (Table 1). There were nine gene modules in the analysis for which the correlation coefficients of genes and metabolites had at least one absolute value greater than 0.6. The module–trait correlation heatmap showed that 19 metabolites were positively correlated with the midnight blue, salmon, purple, and turquoise modules and negatively correlated with the light cyan, magenta, blue, and brown modules (Figure 5, Appendix A).

In the module, the gene with the highest connectivity (k value) was considered the hub gene. By combining known flavonoid and stilbene biosynthesis-related genes and kWithin values, 16 *STS*, 6 *PAL*, and 1 *4CL* hub genes were identified in the midnight blue module (Table 2). We also identified 22 TFs, including *MYB14* from the midnight blue module and *WRKY53* from the salmon module, which are known to be related to flavonoid and stilbene biosynthesis. It should be noted that we further identified several new TFs that might be involved in the regulation of flavonoid and stilbene biosynthesis: *bHLH30*, *ERF3*, and *EFM* from the midnight blue module; *WRKY22*, *MYB36*, *MYB15*, and *WRKY44* from the salmon module; *NAC100*, *ORR26*, and *bHLH87* from the purple module; *ZFP3* and *TCP15* from the magenta module; *BEL1-like protein 3* and *bZIP9* from the blue module; *E2FA* and *bHLH96* from the turquoise module; *E2FE* and *MYB3R-4* from the pink module; and *AGL65* and *WRKY20* from the brown module. These regulators showed expression patterns that were either similar or opposite to most of the structural genes of flavonoid and stilbene biosynthesis, indicating their potential roles in regulating the expression of genes related to the processes of flavonoid and stilbene metabolism (Table 3).

### 2.7. Correlation Analysis between Metabolomic and Transcriptomic Results

The important pathways directly related to the biosynthesis of flavonoids and stilbenes were analyzed, namely flavonoid biosynthesis, phenylpropanoid biosynthesis, and stilbenoid, diarylheptanoid, and gingerol biosynthesis. A total of 91 DEGs were identified as being related to the biosynthesis of flavonoids and stilbenes, with 7 DEGs involved in all three major pathways, 41 DEGs associated with the synthesis of stilbenes, and 56 DEGs associated with the synthesis of flavonoids (Appendix A).

The biosynthetic pathways of flavonoids and stilbenes were further depicted as a heatmap, and the heatmaps of the corresponding metabolite accumulation and gene-expression trends were attached to each step (Figure 6). Among the DCMs, the contents of eight flavonoid metabolites—naringenin, dihydroquercetin, eriodictyol, dihydrokaempferol, prunin, phlorizin chalcone, naringin, and neohesperidin—were higher in Suc7.5-T1 and Suc15-T1 than in Suc7.5-T2 and Suc15-T2, respectively. The flavonoid metabolites naringenin, dihydrokaempferol, prunin, and naringin also had higher contents in Suc30-T2 than in Suc30-T1. These results suggested that, under low sucrose conditions, most flavonoids were at higher levels at 15 d of cultivation than at 30 d. However, under high sucrose conditions, the opposite was true. It is worth noting that phlorizin chalcone was the only identified DCM in Suc15-T1 and Suc15-T2, with its highest content in the former. As for the two stilbene metabolites—resveratrol and pterostilbene—the content of resveratrol was higher in the 15 vs. 30 d group, while pterostilbene content was higher in the 30 d group for all sucrose concentrations. Interestingly, there was a significant negative correlation between *C4H* and p-coumaric acid. Two genes related to *ROMT* (LOC100259140 and LOC100254011) showed a positive correlation with pterostilbene. For the stilbene synthesis-related genes, there were 12 *STS* genes involved in resveratrol synthesis, and the results revealed that the expression levels of most *STS* genes were higher in the T1 vs. T2 group (Figure 6, Appendix A). These results indicated that the expression of most *STS* genes is downregulated with increasing sucrose concentration and cultivation time.

RNA-seq results revealed genes related to flavonoid synthesis, as detected by quantitative real-time PCR (qRT-PCR) (Figure 7). The expression levels of *C4H*, *F3′H*, *F3H*, and *CHI* showed an increasing trend from Suc7.5-T1 to Suc30-T1 and from Suc7.5-T2 to Suc30-T2. Moreover, their expression levels were lower in the Suc7.5-T2 and Suc15-T2 groups, reaching the highest expression level when the sucrose concentration in the culture medium was 30 g/L. Among these, *C4H* and *CHI* had the highest expression levels in Suc30-T1, while *F3′H* and *F3H* had the highest expression levels in Suc30-T2. *F3′5′H* expression exhibited a trend similar to that of other genes related to flavonoid synthesis in the T1 group, while its expression level was highest in Suc15-T2. The gene *VINST1* and its regulatory TF *MYB14* exhibited a V-shaped trend in the T1 group and gradually increased in the T2 group, consistent with the resveratrol contents (Figure 7).

## 3. Discussion

Active ingredient extracts obtained through plant cell culture technology, due to their growth under controlled laboratory conditions, are completely standardized, ensuring the production of high-quality sustainable products at any time. In recent years, plant cell culture technology has enabled the controlled production of bioactive compounds, such as flavonoids and stilbenes, under optimized conditions [22,23,24]. For example, the United States company Phyton successfully produced paclitaxel from yew callus tissue and has been approved by the Food and Drug Administration as a supplier of paclitaxel to Bristol-Myers Squibb. Phyton conducted systematic research on plant cell growth and metabolic product accumulation and achieved good results [25]. Today, most of the active ingredients in the cosmetics industry are obtained through plant cell culture technology by cultivating dedifferentiated callus cultures or dedifferentiated plant cell suspension cultures and then extracting the active ingredient. For example, *Rubus ideaus* hydrosoluble extract has anti-inflammatory activity and DNA-protection and repair properties; *Malus domesticus* whole lysate can revert signs of senescence; trans-resveratrol from *Vitis vinifera* cell culture has antioxidant activity; and a mixture of liposoluble marc extracts (LME) and liposoluble cell culture extract (LCE) of *Vitis vinifera* has hydration and cell-detoxifying activities [22,24,26,27]

The plant callus culture medium provides the necessary nutrients for the induction and growth of callus tissue. It has been suggested that the carbon source is the most basic of these nutrients, with its type and quantity playing a significant role in regulating callus production [28]. In the present study, we used sucrose as the carbon source and evaluated its effect on callus growth by supplementing the MS medium with different sucrose concentrations (10, 20, 30 g/L) and examining the callus after different times of cultivation as well. The grape callus proliferation rate was found to be higher in the MS media with 15 and 30 g/L sucrose compared to 7.5 g/L (Figure 1). In a study using rice seed as the starting material, 20 g/L and 40 g/L sucrose in the medium were considered sugar-deficient and sugar-excessive for both callus induction and growth, while 30 g/L sucrose was found to be the optimal concentration [28]. The most suitable sugar content in the medium could depend on plant species and the specific materials. *Arabidopsis* leaf callus induction and progression were reported to be accelerated with an increase in sugar concentration from 30 g/L to 50 g/L [29], whereas an increase in the fresh weight of non-embryogenic calli induced from coconut anther culture was found to be negatively correlated to increasing sucrose concentrations in the range of 40 to 120 g/L [30]. It was suggested that excessive sugar can lead to the production of more phenolic compounds in cells, resulting in cell death and browning of the callus tissue [31,32]. In our study, we noticed that when the sucrose concentration in the culture medium was increased to 30 g/L, some callus tissue cells turned into white, fluffy tissue with no further growth potential (Figure 1). The optimal concentration of sucrose (15 g/L) provided comparatively better development of yellowish callus tissue compared to the sugar-deficient (7.5 g/L) and excessive (30 g/L) media.

In plant callus medium, sugar not only serves as a carbon source, providing energy for the accumulation of cell biomass, but also acts as a signaling molecule that might regulate the expression of multiple genes in primary and secondary metabolism pathways [33]. In grape cell tissue culture, increased sucrose concentrations have been found to promote cell growth and phenylpropanoid biosynthesis for cells obtained from immature cv. Barbera berries [21]. In our study, there were more DCMs in 7.5 vs. 15 g/L sucrose treatment groups than in 15 vs. 30 g/L groups at 15 d of subculture, whereas more DCMs were identified in 15 vs. 30 g/L sucrose treatment groups at 30 d of subculture. In *Melastoma malabathricum*, sucrose is needed for the enhancement of cell growth and anthocyanin production [34]; in *Hylocereus costaricensis*, a high concentration of sucrose is beneficial for cell mass accumulation, whereas a low concentration of sucrose stimulates the accumulation of protein compounds and antioxidants in the cell suspension culture [35]. 

Grapes contain abundant flavonoids and stilbenes, which are important active substances in many medicines and foods [4,5]. The flavonoid and stilbene biosynthesis pathways have been intensively studied in grapevine [36,37,38,39]. However, there is a lack of information associated with the expression of flavonoids, stilbenes, and their related genes’ expression in grape callus under different sucrose concentrations. In the present study, the grape callus culture technique was under automated control, and all of the growth conditions were fixed except for the concentration of sucrose in the medium and cultivation time, enabling us to analyze the metabolomic and transcriptomic differences in grape callus under these conditions (Figure 1). When the sucrose content is 15 g/L, it is recommended to extract the active substances at 15 d because the contents of most flavonoid and stilbene metabolites were relatively higher under these compared to the other tested conditions; combined with its higher biomass, maximum benefit and time efficiency can be achieved with this cultivation time and sucrose concentration.

In response to sucrose signals, grape callus tissue in this study underwent changes in the accumulation of a series of flavonoid and stilbene compounds, accompanied by regulation of their respective genes’ expression, similar to a study conducted by Lu et al. [40] in grape berries. To comprehensively understand the transcriptomic and metabolomic data obtained in the present study, WGCNA was performed, revealing four co-expression modules/gene networks that were highly correlated with changes in the flavonoid- and stilbene-related compounds (Figure 6). Within these correlations, several key candidate genes were identified as hub genes for their module (Table 3); for example, genes related to flavonoid synthesis, such as *F3H*, *CHI*, *C4H*, and *F3′H*, and genes related to stilbene synthesis, *VINST1*, *STS* (LOC100253166), and *STS-2*. These hub genes might be the major genes controlling flavonoid and stilbene regulation in grape callus under tissue culture conditions. Combined with the qRT-PCR verification results (Figure 7), it was revealed that *C4H*, *F3′H*, *F3H*, and *CHI* have similar expression patterns, and the differential expression of these five genes was positively correlated with the differential contents of the metabolites naringenin and dihydrokaempferol. The expression level of these five genes was highest when the grape callus was cultured with 30 g/L of sucrose for each culture time. The expression pattern of *F3′5′H* was inconsistent with those of the above five genes, which may lead to differences in the accumulation of the metabolites of dihydromyricetin and eriodictyol related to naringenin and dihydrokaempferol. The results of this study indicate that, within the sucrose concentration range of 7.5 to 30 g/L, sucrose upregulates the flavonoid branch of the phenylpropanoid pathway, in accordance with the results of Ferri et al. [21].

In recent years, TFs have been reported to participate in the regulation of flavonoid and stilbene synthesis [10,41,42,43]. In this study, many TFs were identified and might participate in the response of grape callus tissue to sucrose (Appendix A). By using WGCNA for module division and analysis, the positive/negative regulatory factors of flavonoid and stilbene synthesis were identified, including *MYB14*, *MYB15*, *MYBPA1*, *WRKY53*, and *MYC2* [10,44,45]. It might be speculated that TF *MYB14* controls the expression of STS and thereby regulates the synthesis of resveratrol, as proposed in a previous study [10]. In addition, identification of the three candidate TFs—*MYB59*, *WRKY20*, and *MADS8*—provides a new direction for further research on flavonoid and stilbene synthesis. These results suggest that the production and accumulation of flavonoids and stilbenes in grape callus tissue is regulated by a complex network in which TFs may play a significant role.

## 4. Materials and Methods

### 4.1. Plant Materials

Grapevine calli were induced and collected from thin shoot slices of the tissue-cultured variety Mio Red. The collected calli were subcultured every 30 d in MS medium (with 20 g/L sucrose, 2 mg/L TDZ, 0.05 mg/L NAA, and 8 g/L agar, pH 5.8) on Petri dishes in the dark at 25 °C in a growth chamber. 

Sucrose concentrations were set at 7.5, 15, and 30 g/L, with six Petri dishes for each treatment. The callus grown in the media with different sucrose contents were collected at two time points: 15 d (T1) and 30 d (T2). The fresh weight of the callus was recorded, and the samples were frozen in liquid nitrogen and kept at -80 ℃ for further analysis. The collected samples were named Suc7.5-T1/T2, Suc15-T1/T2, and Suc30-T1/T2, respectively, with three biological replicates for each treatment (Appendix A). 

### 4.2. Metabolite Profiling

Metware Biotechnology (Wuhan, China) performed the comprehensive targeted metabolomic analysis and metabolite identification and quantification according to its standard procedures [46,47]. In brief, callus was freeze-dried then crushed using a mixer mill (MM 400, Retsch). The lyophilized powder (50 mg) was extracted with 1.2 mL of 70% methanol solution, centrifuged at 12,000 rpm (15,294 g) for 3 min, and the supernatant was filtered through a 0.22 μm pore size membrane for ultra-performance liquid chromatography–mass spectrometry (UPLC-MS/MS) analysis. Based on the self-constructed Metware database, qualitative analysis of metabolites was conducted using secondary spectrum information, and the triple quadrupole mass spectrometry’s multiple reaction monitoring mode was utilized for quantitative analysis of metabolites [3]. 

Significantly regulated metabolites between groups were determined by variable importance in project (VIP ≥ 1) and fold change (FC) ≥ 2 or ≤ 0.5. VIP values were extracted from orthogonal partial least square discriminate analysis (OPLS-DA) results, which also contained score plots and permutation plots generated using the R package Metabo Analyst R (MetaboAnalyst 5.0). The data were log-transformed (log2) and mean-centered before OPLS-DA. Identified metabolites were annotated using the Kyoto Encyclopedia of Genes and Genomes (KEGG) compound databases and the Metware database and were then mapped to the KEGG Pathway database. DCMs were screened with FC ≥ 2 and VIP ≥ 1.

### 4.3. Transcriptome Analysis

Grapevine calli cultured with different sucrose concentrations were separately harvested and finely ground in liquid nitrogen. Total RNA was isolated from 200 mg of the ground powder using the CTAB method [48]. After quality checking, 1 μg of RNA was used for library construction. Sequencing was carried out with the Illumina Novaseq6000 Platform (Metware Biotechnology). The raw data were filtered using fastp v0.19.3, and then the clean reads were mapped to the grape genome (https://www.ncbi.nlm.nih.gov/genome/?term=txid29760/ accessed on 24 September 2024) using HISAT v2.1. We used Feature Counts v1.6.2 to calculate the gene alignment and FPKM (mapped fragments of transcript/total count of mapped fragments [millions] x length of transcript [kb]). DEGs between any two groups were screened using the DESeq2 software (v1.42.1) under the conditions of |log2 (FC)| ≥ 1 and false discovery rate < 0.05. The identified DEGs were then further analyzed for KEGG enrichment.

### 4.4. Association Analysis of Metabolome and Transcriptome Data

The DEGs were used to generate a co-expression network module using the WGCNA package in R (R version 3.5.1) after eliminating undetectable or relatively low expression genes (FPKM < 10) [49]. The automatic network construction function (blockwise modules) with default parameters was employed to obtain co-expression modules using the Metware Cloud, a free online platform for data analysis (https://cloud.metware.cn/ accessed on 24 September 2024). Initial clustering was merged on characteristic genes. Feature gene values were calculated for each module and used to search for associations with metabolites related to flavonoid and stilbene synthesis.

### 4.5. qRT-PCR Analysis

Nine DEGs were selected for qRT-PCR to validate their expression patterns. The specific primers were designed by Primer 5.0 software (Appendix A), and PCR amplification was conducted using the QuantStudio 6 Flex real-time PCR system (Thermo Fisher Scientific, Foster City, USA) with the TransStart Top Green qPCR SuperMixkit (Q711-02, Vazyme, Nanjing, China). The PCR system consisted of 5 μL of Mix, 0.25 μL each of forward and reverse primers, 2 μL of cDNA, and 2.5 μL of deionized water. The PCR program was as follows: pre-denaturation at 95 °C for 10 min, denaturation at 95 °C for 30 s, annealing at 58 °C for 30 s, and extension at 72 °C for 30 s, for a total of 45 cycles. The relative expression levels were calculated using 2^−ΔΔCt^, and the transcript level of *Actin* (LOC100232866) was used for internal normalization. Three biological replicates and three measurements for each replicate were performed under identical conditions.

## 5. Conclusions

In this study, metabolomics and transcriptomics analyses using UPLC-MS/MS and RNA-seq techniques were conducted to understand the production of flavonoids and stilbenes in grape callus under the influence of sucrose concentration and cultivation time. The main results demonstrated that sucrose concentration and cultivation time significantly impact the accumulation of flavonoids and stilbenes in grape callus. Comparative analyses between gene expression and metabolite accumulation in the flavonoid pathway revealed that five genes (*C4H*, *F3′H*, *F3H*, and *CHI*) play key roles in regulating flavonoid biosynthesis. In addition, novel regulatory candidates were identified (*MYB59*, *WRKY20,* and *MADS8*), which might play an important role in flavonoid and stilbene biosynthesis in grape callus. In summary, this study not only provides supporting data for the investigation of flavonoid and stilbene metabolite production in grape callus but also offers new information to understand the regulation of flavonoid and stilbene synthesis in the callus. Our results will serve to optimize grape callus cultures for the production of flavonoids and stilbenes to be used in medical, health, and cosmetic products.

## Figures and Tables

**Figure 1 ijms-25-10398-f001:**
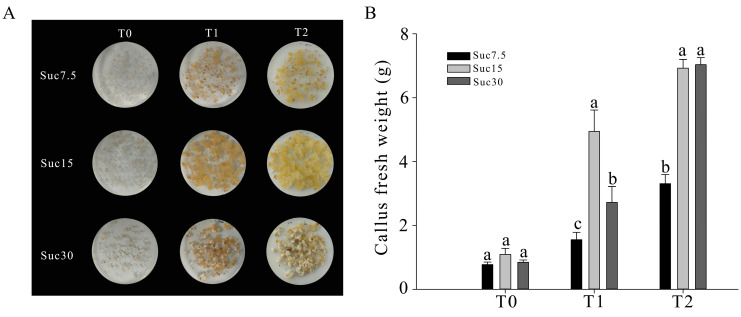
Phenotype (**A**) and fresh weight (**B**) of grape callus tissues grown on media with different sucrose concentrations for different times in culture (in days). Small letters (a, b, and c) represent a statistically significant difference (*p* < 0.05) analyzed using Student’s *t*-test.

**Figure 2 ijms-25-10398-f002:**
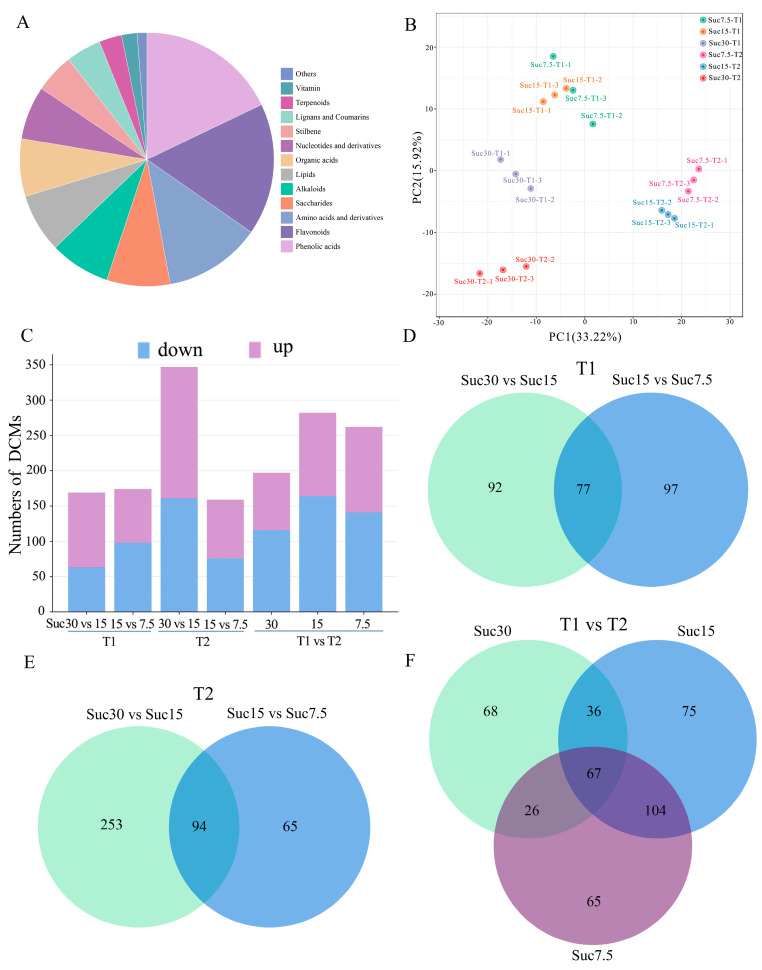
Multivariate statistical analysis of metabolome data from grape callus tissues. (**A**) Classification of metabolites with differential contents (DCMs). (**B**) PCA of the metabolic results. (**C**) Numbers of metabolites with differences in relative abundance in different comparison groups (T1, T2, and T1 vs. T2); purple and blue columns represent the numbers of metabolites with increased or decreased relative abundance, respectively. Venn diagram showing the number of overlapping and unique DCMs in the comparison groups T1 (**D**), T2 (**E**), and T1 vs. T2 (**F**).

**Figure 3 ijms-25-10398-f003:**
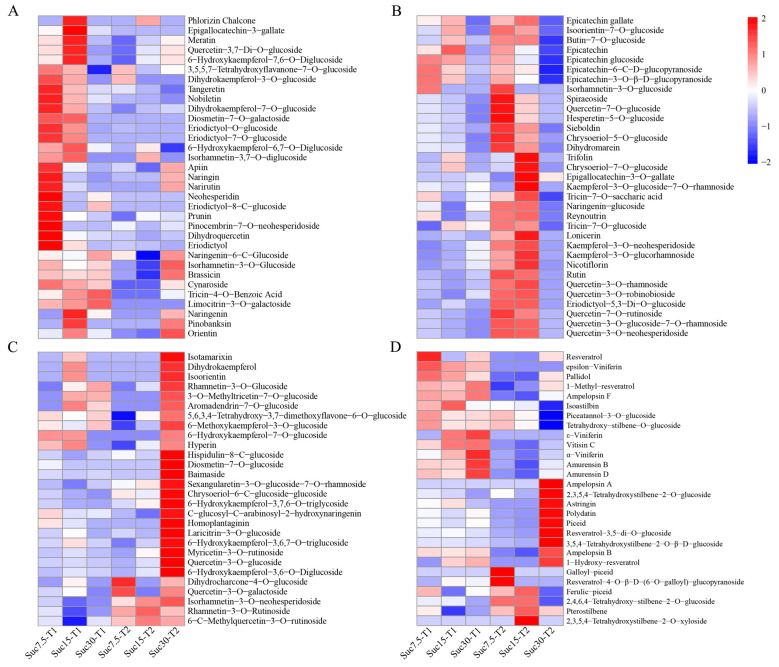
Analysis of the relative concentrations of flavonoids and stilbenes. Heatmap analysis of flavonoids with high expression in groups Suc7.5-T1 and Suc15-T1 (**A**), Suc7.5-T2 and Suc15-T2 (**B**), and Suc30-T2 (**C**). (**D**) Heatmap analysis of stilbenes.

**Figure 4 ijms-25-10398-f004:**
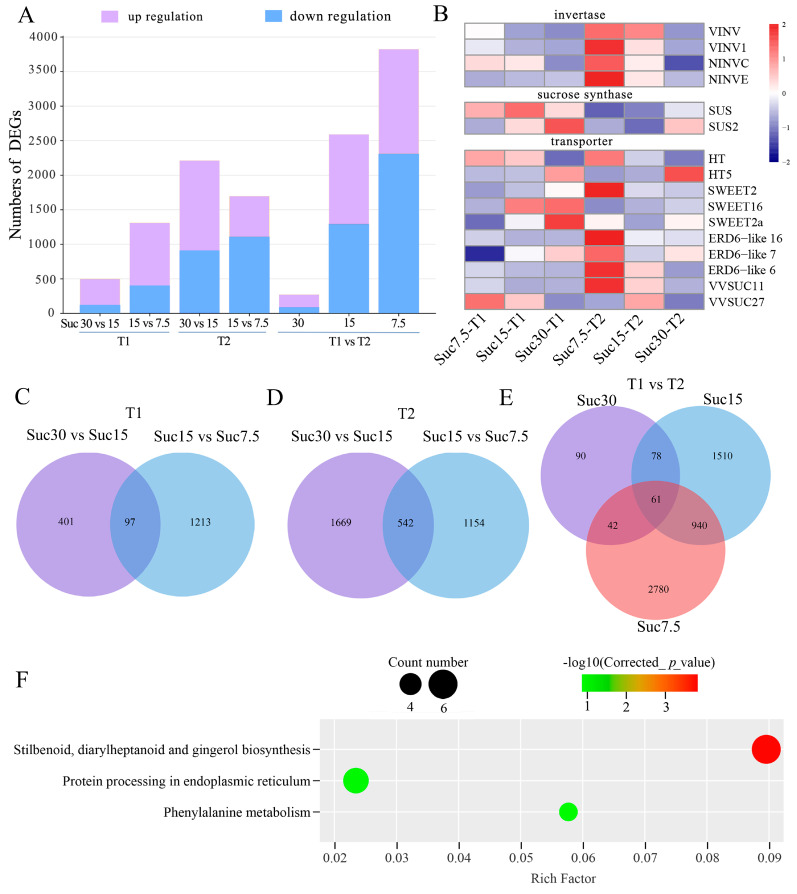
Multivariate statistical analysis of transcriptomic data among grape callus tissues. (**A**) Numbers of differentially expressed genes (DEGs) in different comparison groups (T1, T2, and T1 vs. T2); purple and blue columns represent the numbers of genes with increased or decreased rela-tive expression, respectively. (**B**) Heatmap analysis of the 16 DEGs related to invertase, sucrose synthesis, and sugar transport. Color shading from blue to red in the heatmap indicates the rela-tive expression levels of the genes, ranging from low to high. (**C**–**E**) Venn diagrams showing the overlapping and unique DEGs in the comparison groups T1 (**C**), T2 (**D**), and T1 vs. T2 (**E**). (**F**) KEGG enrichment analysis of DEGs in the comparison T1 vs. T2.

**Figure 5 ijms-25-10398-f005:**
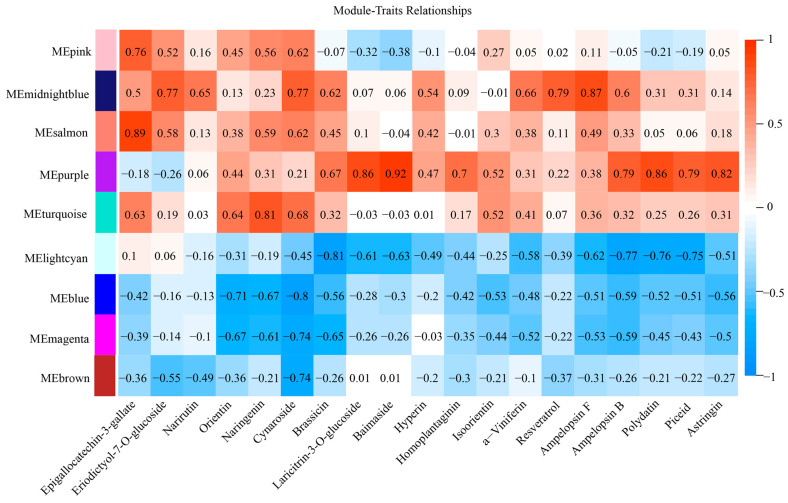
Module–trait association of 12 flavonoid and 7 stilbene metabolites with 9 gene modules. Each column corresponds to a specific metabolite, and each row corresponds to a module. The values in each cell indicate the correlation coefficient between the module and the metabolite, and the color strength is related to the *p*-value, with a high positive correlation indicated by dark red and a strong negative correlation indicated by dark blue.

**Figure 6 ijms-25-10398-f006:**
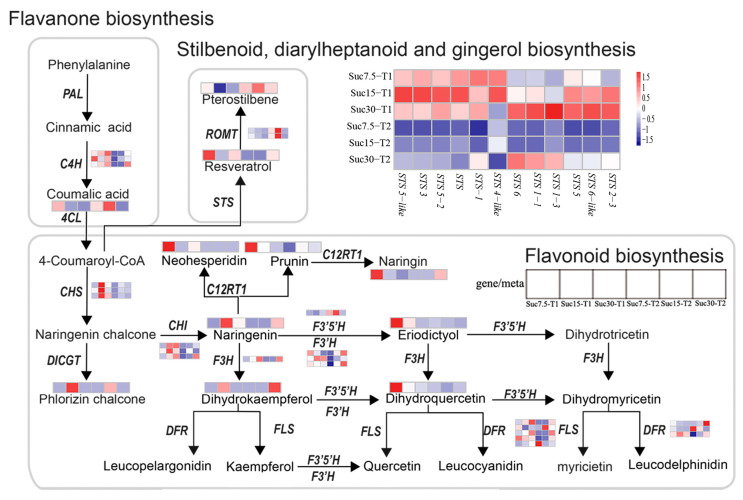
Differentially expressed genes (DEGs) and differential content metabolites (DCMs) involved in flavonoid and stilbene biosynthesis.

**Figure 7 ijms-25-10398-f007:**
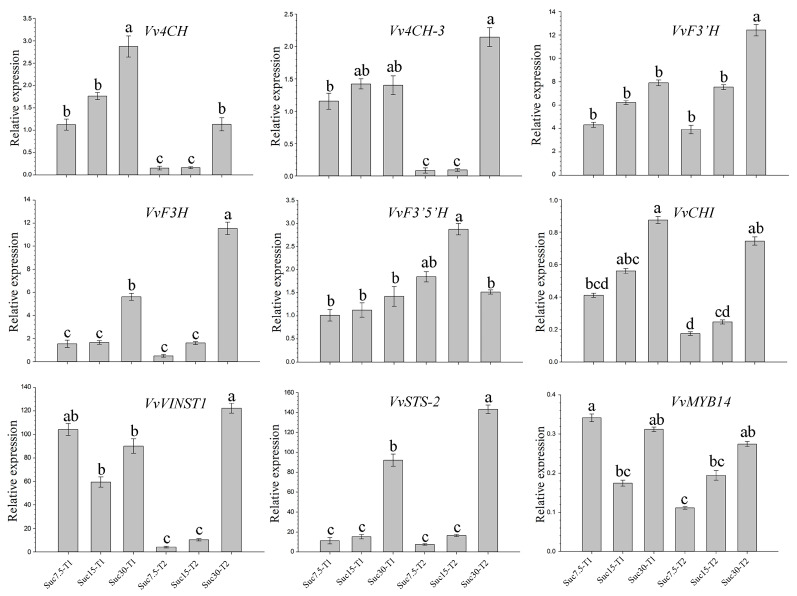
qRT-PCR verification of the expression levels of key genes involved in the flavonoid and stilbene biosynthesis pathways in grape callus tissue samples. Small letters (a, b, c, and d) represent a statistically significant difference (*p* < 0.05) analyzed using Student’s *t*-test.

**Table 1 ijms-25-10398-t001:** Levels of 19 flavonoid and stilbene metabolites with differential contents (DCMs) in different treatment groups.

Compounds	Suc7.5-T1	Suc15-T1	Suc30-T1	Suc7.5-T2	Suc15-T2	Suc30-T2
**Flavonoids**						
Epigallocatechin-3-gallate	9.72 × 10^4^	3.97 × 10^5^	4.91 × 10^4^	ND	ND	ND
Eriodictyol-7-O-glucoside	1.57× 10^5^	1.14 × 10^5^	ND	ND	ND	ND
Narirutin	1.03 × 10^5^	1.75 × 10^4^	ND	ND	ND	5.94 × 10^4^
Orientin	1.92 × 10^4^	4.28 × 10^4^	1.94 × 10^4^	4.40 × 10^3^	ND	5.01 × 10^4^
Naringenin	ND	3.85 × 10^4^	1.37 × 10^4^	ND	ND	2.15 × 10^4^
Cynaroside	4.20 × 10^5^	3.47 × 10^5^	3.01 × 10^5^	ND	ND	2.65 × 10^5^
Brassicin	3.11 × 10^5^	2.84 × 10^5^	3.04 × 10^5^	1.28 × 10^5^	ND	4.04 × 10^5^
Laricitrin-3-O-glucoside	9.94 × 10^5^	1.15 × 10^6^	8.90 × 10^5^	1.02 × 10^6^	9.28 × 10^5^	1.94 × 10^6^
Baimaside	2.19 × 10^5^	2.00 × 10^5^	2.14 × 10^5^	1.91 × 10^5^	1.94 × 10^5^	6.43 × 10^5^
Hyperin	8.37 × 10^5^	9.22 × 10^5^	5.44 × 10^5^	4.31 × 10^5^	6.92 × 10^5^	9.18 × 10^5^
Homoplantaginin	9.10 × 10^4^	4.37 × 10^4^	ND	ND	ND	2.77 × 10^5^
Isoorientin	ND	2.60 × 10^4^	ND	ND	ND	3.93 × 10^4^
**Stilbenes**						
α-Viniferin	1.24 × 10^6^	1.85 × 10^6^	2.87 × 10^6^	6.13 × 10^5^	2.27 × 10^5^	9.20 × 10^5^
Resveratrol	2.32 × 10^7^	4.22 × 10^6^	1.21 × 10^7^	1.26 × 10^6^	9.28 × 10^5^	1.12 × 10^7^
Ampelopsin F	3.55 × 10^5^	3.27 × 10^5^	4.02 × 10^5^	6.67 × 10^4^	2.73 × 10^4^	2.23 × 10^5^
Ampelopsin B	1.05 × 10^7^	1.02 × 10^7^	1.07 × 10^7^	4.16 × 10^6^	2.59 × 10^6^	1.59 × 10^7^
Polydatin	4.08 × 10^6^	3.21 × 10^6^	4.40 × 10^6^	1.89 × 10^6^	1.63 × 10^6^	1.01 × 10^7^
Piceid	2.57 × 10^7^	2.15 × 10^7^	2.71 × 10^7^	1.54 × 10^7^	1.28 × 10^7^	5.67 × 10^7^
Astringin	1.94 × 10^4^	2.68 × 10^4^	7.76 × 10^3^	ND	ND	6.60 × 10^4^

ND: not detected.

**Table 2 ijms-25-10398-t002:** Hub structural genes related to flavonoid and stilbene biosynthesis from module midnight blue in WGCNA analysis.

Gene ID	Name	kTotal	kWithin
LOC100258294	*STS 1-like*	194.23	47.26
VINST1	*STS-1*	375.33	43.10
LOC100246283	*STS 5-1*	143.34	41.90
LOC100855299	*STS 2-1*	244.78	41.59
LOC100242903	*STS 4-1*	150.72	40.32
LOC100261661	*STS 1-2*	418.83	40.13
LOC100245181	*STS 5*	551.28	36.40
LOC100853590	*STS 4-like-3*	135.31	36.17
LOC100853745	*STS 4-like-5*	208.68	35.54
STS	*STS*	259.87	35.04
LOC100241164	*STS 6-like*	495.22	32.88
LOC100853820	*STS 4-like-6*	126.42	32.82
LOC100266562	*STS 2-4*	200.09	31.27
LOC109121404	*STS 4-like-7*	131.97	30.98
LOC100240862	*STS 2-2*	155.77	30.54
LOC100853675	*STS 4*	194.21	30.42
LOC100233012	*PAL G1*	335.58	42.80
LOC100855356	*PAL-1*	225.87	42.47
LOC104881994	*PAL1-like*	256.86	41.46
LOC100266593	*PAL-2*	318.58	39.04
LOC100240904	*PAL-like-1*	244.57	36.32
LOC100256293	*PAL-like-2*	268.50	32.90
LOC100254698	*4CL-2*	249.92	39.96

**Table 3 ijms-25-10398-t003:** Hub transcription factors related to flavonoid and stilbene biosynthesis from key modules in the WGCNA analysis.

Gene ID	Name	kTotal	kWithin	Module Colors
LOC100256680	*bHLH30*	210.06	32.46	midnight blue
MYB14	*MYB14*	569.75	32.46	midnight blue
LOC100240970	*ERF3*	237.17	28.97	midnight blue
LOC100267475	*EFM*	511.43	26.98	midnight blue
LOC100251862	*WRKY22*	390.74	48.26	salmon
LOC100243880	*MYB36*	270.73	47.09	salmon
LOC100254518	*MYB15*	378.42	45.66	salmon
LOC100256922	*WRKY53*	273.87	45.06	salmon
LOC100256868	*WRKY44*	377.95	43.76	salmon
LOC100266459	*NAC100*	685.91	196.90	purple
LOC104882333	*ORR26*	369.73	184.61	purple
LOC100251455	*bHLH87*	711.39	178.60	purple
LOC100245739	*ZFP3*	722.76	221.87	magenta
LOC100245013	*TCP15*	966.92	221.44	magenta
LOC100257488	*BEL1-like protein 3*	1816.60	865.65	blue
LOC100264680	*bZIP9*	1764.46	857.84	blue
LOC100249486	*E2FA*	1581.44	623.29	turquoise
LOC100261443	*bHLH96*	1223.52	581.65	turquoise
LOC100241701	*E2FE*	809.73	295.98	pink
LOC100261001	*MYB3R-4*	752.70	277.57	pink
LOC100250157	*AGL65*	1158.16	525.52	brown
WRKY20	*WRKY20*	1192.95	508.63	brown

## Data Availability

The transcriptome data of all samples have been uploaded to the NCBI Sequencing Read Archive (SRA) database (https://www.ncbi.nlm.nih.gov/sra/PRJNA1144781/ accessed on 24 September 2024) under the accession number PRJNA1144781. The data that support the findings of this study are available from the corresponding author upon reasonable request.

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
