# Peer review of "Metabolome and Transcriptome Joint Analysis Reveals That Different Sucrose Levels Regulate the Production of Flavonoids and Stilbenes in Grape Callus Culture"

_ijms, 2024, doi:10.3390/ijms251910398_

Round 1

Reviewer 1 Report

Comments and Suggestions for Authors

Using a combination of metabolomics and transcriptomics analysis, the authors checked the effects of sucrose concentration on the accumulation of flavonoids and stilbenes in grape callus cells.  Moreover, they have identified several important genes. This work provides meaningful information about the grape callus culture and deepens our understanding of the flavonoids and stilbenes synthesis in grapes. However, some contents need to be adjusted and updated.

1. There may be huge differences in the flavonoids and stilbenes of different grape varieties, and the authors need to add some recent research in the Introduction. 

2. In Figure 7, the authors should thoroughly verify the significance analysis. For example, in the VvVINST1 expression, it was easy to see that the Suc30-T1 set has a significant difference with Suc7.5-T2 and Suc15-T2.

Comments on the Quality of English Language

1. The authors need to specify the grape variety of the callus in "Abstract". In lines 12-13, the sentence "and the corresponding callus were analyzed using metabolome and transcriptome techniques." was hard to understand.

2. Line 39 has two repetitions of the word "flavanols".

3. The legend of Figure 4 needs to be congruent with the legends of other figures.

4. Line 453, "

Reviewer 2 Report

Comments and Suggestions for Authors

Very interesting study offering valuable insights into the metabolic regulation of secondary metabolites in grape callus. The experimental approach is thorough and well-executed, and the findings have important practical applications. However, there are many weaknesses too. First of all, there is plenty of minor grammatical errors. The paper would definitely benefit from stronger writing, clearer structure, and more focus on biological interpretation and comparison with existing research. Please find below the comments, corrections and suggestions that need to be addressed:

Line 2. The first part of the title is wrong. The verb “reveals” should agree with “analysis” which is singular. So, “reveals” instead of “reveal” is the correct form. Also, the phrase following the word “reveal” doesn’t really make sense. You need to add “that” or “how” after the word “reveal” to make sense. For example, a correct title could be: Metabolome and transcriptome joint analysis reveals that different sucrose levels regulate the production of flavonoids and stilbenes in grape callus culture”. An alternative could be: “Joint metabolome and transcriptome analysis reveals how sucrose levels regulate flavonoid and stilbene production in grape callus culture”.

Lines 10-34. The abstract is overly long and is not written very well. Some elements are redundant. I recommend you re-write a clearer, shorter and more comprehensive Abstract. Find below some specific examples of inadequate writing:

Lines 10-11.  “of the secondary metabolites”. The word “the” is not needed here.

Lines 18-19. The phrase “At 15 d, there 1310 differentially expressed genes (DEGs) were screened out” doesn’t make sense. “There” is not needed and ruins the structure of the sentence. Exactly the same happens in LINES 14 and 30. Please remove the word “there”.

Lines 21-22: “At 30 d, there were 1696 and 1711 DEGs were screened out, respectively.” is repetitive and could be clarified as “At 30 days, 1696 and 1711 DEGs were identified, respectively.”

Line 24. “Higher expressed” could be substituted with “more highly expressed” for a clearer meaning. Or the whole sentence could be re-written.

Lines 37-83: The introduction provides necessary background but can be reorganized for better flow. For example, more emphasis should be placed on the significance of the study, the gap in current knowledge, and why metabolome and transcriptome analyses are appropriate. Some sections feel a bit redundant and can be streamlined, especially regarding the benefits of flavonoids and stilbenes.

Lines 41-43: This sentence could be modified to “Studies have demonstrated that flavonoids and stilbenes possess antioxidant, anticancer, antibacterial, and anti-inflammatory properties, among other health benefits.”

Line 46-47: Although a citation is provided in line 49, one more reference could be provided in lines 46-47, as “The biosynthesis of both flavonoids and stilbenes starts from the phenylpropane pathway” is a scientific statement

Line 49. As the sentence begins with “Subsequently” the word “then” in line 49 is not needed. Actually, this is a pleonasm (repetitive phrase).

Lines 57-58. The phrase “from specific tissue parts of plants” is not structured very well. Did you mean “from specific parts of plant tissues”?

Line 60: A gap (space) is needed between the word “callus” and [17].

Lines 84-295. The results section is dense with data, figures and tables. Although there are subsections, it could be easier to follow if organized into more distinct subsections (i.e., clearer separation of metabolome, transcriptome, and co-expression network results). Consider adding transitional sentences to guide the reader between different sections of the results. Some figures are a little hard to read. Some parts of the results section contain a lot of data (e.g., percentage growth rates, DEGs, DEMs), but the biological interpretation of these findings is sometimes lacking.

Line 97. Callus is a singular form, so “was” should be used instead of “were”.

Lines 100-102. The letter d is used as an abbreviation throughout the manuscript but is nowhere explained. I suppose that’s fine, as it is a very common abbreviation for days. It is also a common abbreviation for density, so I am not sure about that. However, I recommend you change the second line of the legend of figure 2 (line 102) to: “Concentration and culture time (in days)”.

Lines 123-125: “Flavonoids were also the most abundant differential metabolites in the 'T1' and 'T2' comparison groups...” Rephrase for clarity: “Among the identified DEMs, flavonoids represented the most abundant category in both 'T1' and 'T2' groups, accounting for 36.36% and 18.09%, respectively.”

Lines 236-291. The discussion section explains the findings but could better tie the results back to the research question. Specifically, more focus could be given to how sucrose concentration affects flavonoid and stilbene production. The discussion lacks a critical comparison of the findings with existing literature. Highlight how your study supports or contradicts previous research.

Lines 297-299: This sentence introduces new information about sustainability, which doesn't seem directly relevant to the findings. You could streamline this and focus more on the findings: “Plant cell culture technology enables the controlled production of bioactive compounds, such as flavonoids and stilbenes, under optimized conditions.”

Lines 335-340: You can expand this to make a direct comparison with your findings.

Line 342. Why are you using a past tense? “were” doesn’t really make sense.

Lines 392-456. More references could be added in some parts of the section.

Lines 394-397. The entire sentence needs to be revised and written in a clearer way.

Lines 425-427. Clarify that this ground powder comes from the harvested callus and give specifics about the RNA extraction kit (e.g., manufacturer, kit name). Provide more information about the method or/and citation.

Line 438. Specify the version of the WGCNA package used and add details like the threshold values or parameters for module identification 

Lines 458-470. The conclusion is relatively solid but should include a stronger statement about the implications of the research for future studies or applications.

Comments on the Quality of English Language

There are minor language - grammar errors throughout the manuscript. Even in the title. In some sections, the level of English is not acceptable for publication, while in other parts it is better.

Round 2

Reviewer 2 Report

Comments and Suggestions for Authors

The abstract is still long and there are some minor grammatical mistakes too. The discussion section could also be enriched with more comparisons to the existing literature. However, changes and corrections have been applied properly. The overall quality of the manuscript is much higher now.

Comments on the Quality of English Language

The use of the English language has been improved
